# Oxcarbazepine for Behavioral Disorders after Brain Injury: Factors Influencing Efficacy

**DOI:** 10.3390/brainsci11070949

**Published:** 2021-07-19

**Authors:** Marco Pozzi, Paolo Avantaggiato, Valentina Pastore, Carla Carnovale, Emilio Clementi, Sandra Strazzer

**Affiliations:** 1Scientific Institute IRCCS Eugenio Medea, 23842 Bosisio Parini, Italy; marco.pozzi@lanostrafamiglia.it (M.P.); avantaggiato74@yahoo.it (P.A.); valentina.pastore@lanostrafamiglia.it (V.P.); emilio.clementi@unimi.it (E.C.); 2Unit of Clinical Pharmacology, Department of Biomedical and Clinical Sciences L. Sacco, “Luigi Sacco” University Hospital, Università di Milano, 20157 Milan, Italy; carla.carnovale@unimi.it

**Keywords:** oxcarbazepine, psychomotor agitation, severe acquired brain injury, neurological rehabilitation

## Abstract

Carbamazepine and oxcarbazepine are used for behavioral disorders following organic diseases. After severe acquired brain injury, patients may develop frontal symptoms. In our neurological rehabilitation routine, oxcarbazepine is used for better safety over carbamazepine, although its efficacy is not clarified. We aimed to improve knowledge on this use of oxcarbazepine, by probing clinical factors associated with response. We retrospectively examined the clinical records of our patients, collecting clinical variables and outcomes of efficacy, both clinician-rated and caregiver/self-rated. We described the distribution of clinical variables and examined their associations via logistic regressions. Patients in our cohort were predominantly pediatric, with frontal lobe damage and irritable/reactive. With an oxcarbazepine median dose of 975 mg, almost half of patients improved. We found several clinical factors associated with clinician-rated efficacy: absence of frontal damage and absence of irritability/reactivity symptoms; clinical factors associated with caregivers/patients-rated efficacy were: higher DRS score at baseline and higher patient age. In this retrospective study, we observed that oxcarbazepine was differentially efficacious in patients with specific characteristics. Our study could not examine drug therapy separately from neuropsychological therapy, nor the influence of dose. Our associative results should be verified experimentally, also assessing causality and establishing dose-related efficacy and safety.

## 1. Introduction

Following severe acquired brain injury (sABI), many patients continue to experience major cognitive and behavior problems after physical recovery; these in turn affect rehabilitation negatively and have social consequences [1].

In cases that require pharmacological treatment, options include psychiatric drugs, from antipsychotics to mood stabilizing anticonvulsant and sedatives, used off-label and hence not thoroughly tested for efficacy and safety [2].

Among these drugs, anticonvulsants have gained momentum as alternatives and adjuncts to antipsychotics and lithium in the treatment of behavior disorders due to affective/schizoaffective disorders. Anticonvulsants can have a lithium-like clinical efficacy profile on behavior, yet they carry less risk for adverse effects, leading to greater manageability and patient compliance [3,4]. Most notably, carbamazepine was effective at improving endogenous mania in patients with organic psychoses and appeared more effective in patients with mixed mania, rapid cycling, and “non-classical” bipolar disorders [4,5]. Carbamazepine appears to be a drug of choice in the treatment of personality disorder [6,7] especially when symptoms of aggressive or impulsive behavior are present [8,9]. Carbamazepine has been used successfully in small cohorts of patients with agitation due to brain injury [10,11], and, more recently, a treatment guideline for behavioral disorders after traumatic brain injury indicated carbamazepine and valproic acid as first line medications over adrenergic antagonists and antipsychotics [12]. The clinical utility of carbamazepine is also limited by multiple drug interactions and adverse reactions. Regarding interactions, carbamazepine is a strong inducer of CYP3A4 and an inducer of glucuronyltransferase [13]. In the clinical setting of brain injury rehabilitation, typical concomitant therapies include drugs with a narrow therapeutic index metabolized by the two above enzymes, such as antibacterial chemotherapies, oral anticoagulants, drugs used for cardiovascular protection, other anticonvulsants, and glucocorticoids. Adverse reactions to carbamazepine itself include hyponatremia, cognitive impairment, hepatic toxicity, anemia, and immune system activation with potentially fatal outcomes [14]. Oxcarbazepine [15] is a derivative of carbamazepine that has the same main mechanism of action, as they both prolong the inactivation phase of voltage-gated sodium channels. However, oxcarbazepine is a weak inducer of CYP3A4 and of glucuronyltransferase, such that the concomitant therapies typical of patients with sABI would not need a dose adjustment [16], which represents a considerable added therapeutic value. The adverse reactions profile of carbamazepine and oxcarbazepine is almost identical. Better tolerance of oxcarbazepine stems from its increased manageability when drug combinations are frequent and inevitable, as in the case of patients with sABI. A lessened impact on the alterations of plasma levels of other drugs directly leads to reduced occurrence of adverse reactions and indirectly translates into better tolerance of oxcarbazepine and easier clinical management. Several studies suggest that oxcarbazepine has effects similar to those of carbamazepine in bipolar disorders [17]. In small active comparative studies, oxcarbazepine was as effective as lithium, haloperidol, and valproate in the treatment of acute mania [18,19]. In recent years, several authors have reported on cases with a long history of bipolar II disorder and violent behavior that showed an improvement in mood stabilization following treatment with oxcarbazepine [20,21,22]. Others have described cases of reduction in acute manic symptoms after oxcarbazepine [23]. There are also negative reports, as oxcarbazepine has been tested without success in a trial on patients with aggression due to dementia [24], and a recent meta-analysis on agitation due to dementia found no efficacy of oxcarbazepine [25]. It is important to note that patients examined in the studies above were not comparable to young or adult patients who incurred brain injury; thus results, both negative and positive, should be considered provisional. More recently, a Cochrane Review has concluded that, although numerous drugs have been tried in the management of aggressive behavior in acquired brain injury, no firm evidence of their efficacy has been found, except for propranolol [7]. Oxcarbazepine and valproate efficacy is likewise unsupported by a sufficient level of evidence [7]. A recent randomized controlled trial demonstrated that carbamazepine was efficacious in controlling aggression after traumatic brain injury [26], adding higher quality evidence to the field. In this context of unclear but possible efficacy, it is most important to choose drugs with few side effects and monitor their effects over time. Studies with a pediatric focus should also be conducted, as there is currently no evidence base on the use of carbamazepine and its derivatives for the control of behavior after brain injury in pediatric patients.

In our clinical practice we regularly use oxcarbazepine as pharmacotherapy for behavioral disorders after brain injury [27], making it possible to analyze its use on a fair number of patients. The objectives of this study were: (1) to describe the efficacy of oxcarbazepine on the containment of symptoms of conduct disorder and oppositional defiant disorder in our clinical population; (2) to verify whether oxcarbazepine efficacy was associated with differences in any clinical variables. The results of our analysis provide indications that distinct groups of patients may benefit more or less from oxcarbazepine; we described also its adverse effects.

## 2. Materials and Methods

### 2.1. Data Source and Extraction

This retrospective study evaluated for inclusion the clinical records of all patients admitted to the sABI unit of our rehabilitation institute during the last 20 years. Inclusion criteria for this study were: (1) having incurred sABI (GCS < 8); (2) being admitted to our institute for rehabilitation; (3) having results from magnetic resonance imaging performed prior to admittance to rehabilitation; (4) having a diagnosis of behavioral disorders formulated by staff neuropsychologists. This diagnosis followed the presence of the criteria that by the DSM-IV-TR lead to the diagnosis of conduct disorder or oppositional-defiant disorder. However, due to the presence of a brain injury not fully recovered, a formal diagnosis of conduct disorder or oppositional-defiant disorder was not made until discharge; the details of behavioral disorders were reported on clinical files and monitored; (5) having a Clinical Global Impression-Improvement (CGI-I) rating of behavioral symptoms (including but not limited to irritability, agitation, escaping, disruption, aggression), by which improvement is defined as CGI-I less than 3 on a scale of 1 (extremely improved) to 7 (extremely worsened); (6) having the referring physician opt for pharmacological treatment of behavioral disorders with oxcarbazepine. Exclusion criteria comprised: (1) having a psychiatric diagnosis before incurring sABI; (2) using psychotropic drugs before incurring sABI; (3) using oxcarbazepine as an anticonvulsant. Once in rehabilitation, all patients received the same standard treatments, as required by their injury (neurological and physical, kinesiotherapy, speech therapy). Psychological support and therapy are included for patients with a level of functioning sufficient to allow interaction; in the present group of patients, cognitive/behavioral psychological therapy was administered to everyone, following clinical protocols previously published [28]. For all included patients, we recorded the following variables: sex (m/f); date of birth; date of admittance to rehabilitation; brain injury etiology (trauma, stroke, anoxic brain damage, other); GCS score (the earliest available); GOS score at admittance to rehabilitation; duration of coma (days). From results of magnetic resonance imaging (MRI), we recorded the presence (yes/no) of damage to areas that may be involved with behavior and impulse control: frontal lobe, callous body, thalamus, or diffuse axonal injury. From results of psychological assessments (including, where applicable: Wechsler Intelligence Scale for Children, Leiter-R, Vineland Adaptive Behavior Scales, Child Behavior Checklists, Adult Behavior Checklist), we recorded the presence (yes/no) of behavioral disorders, subdividing them by distinct types of internalizing or externalizing disorders, in partial accordance with previous work [29]: oppositional behavior or non-compliance; hyperactivity or impulsivity; uninhibition or recklessness; confabulation or perseveration; irritability or over-reaction; aggression; mood disorders. We also recorded the use of concomitant drugs, classifying them as antidepressant or nootropic; antipsychotic or mood stabilizing; antispastic. We registered the dose of oxcarbazepine used for maintenance therapy and the occurrence and type of adverse drug reactions (ADRs) to oxcarbazepine, with details on actions taken to manage them and their outcomes. We collected the scores of the Disability Rating Scale (DRS) [30] and Functional Independence Measure (FIM—Subscale social function) [31] at admittance and at discharge from rehabilitation. For patients who had it administered, we collected scores of two behavior scoring scales. The Agitated Behavior Scale (ABS) rated by caregivers [32] and the Strengths and Difficulties questionnaire (SDQ) self-rated [33] were collected (where available) at two time points, before and after oxcarbazepine administration, which corresponded to admittance and discharge from rehabilitation. If two ABS or SDQ measures were not available, the patient was only evaluated following clinician-rated CGI-I.

### 2.2. Data Analysis

Variables were tested for normality by Kolmogorov–Smirnov or Wilcoxon Tests, as relevant. We described continuous variables as means with standard deviation (if normally distributed), or as medians with first and third quartile; categorical variables were reported as numbers and percentages. The cohort was described subdividing it between patients who responded or not. Variables were compared between responders and non-responders for descriptive purposes, by means of chi squared or the Kruskal–Wallis test or ANOVA as applicable. To define responders and non-responders, we used the main study outcome, i.e., CGI-I scale rated by clinicians 1 to 2 indicated responders, and scores of 3 or more indicated non-responders. Regarding caregiver- and patient-rated scales (ABS and SDQ), we calculated the Reliable Change Index (RCI) [34] using normative data [35,36], and used the RCI as a secondary study outcome. Patients who had a RCI < −1.96 were considered as responders. To assess the association between clinical variables and the clinician-rated response to oxcarbazepine on the CGI-I, we built a logistic regression model with a stepwise approach (*p*-in < 0.05; *p*-out > 0.1). The overall fit, adjusted R^2^, and lack-of-fit were reported; significant predictors were expressed with *p*-values and with odds ratios (OR). To assess the association between clinical variables and caregiver/patient-rated responses to oxcarbazepine on the ABS/SDQ, we built a linear regression model with a stepwise approach (*p*-in < 0.05; *p*-out > 0.1). The overall fit and adjusted R^2^ were reported; significant predictors were expressed with *p*-values and beta parameters. The full list of variables tested for associations was: Glasgow Coma Score (GCS) at injury; duration of coma (days); Glasgow Outcome Score (GOS) at admittance to rehabilitation; Disability Rating Scale (DRS) at admittance to rehabilitation and at discharge; Functional Independence Measure (FIM) at admittance to rehabilitation and at discharge; age at start of oxcarbazepine administration (years); oxcarbazepine titrated dose administered (mg/day); damage evidenced by MRI to the frontal lobe/callous body/thalamus/diffuse axonal damage (each yes/no); presence at neuropsychological evaluation of opposition, noncompliance/of hyperactivity, impulsivity/of uninhibition, recklessness/of confabulation, perseveration/of irritability, reactivity/of aggression/of disordered mood (each yes/no); concomitant use of antidepressant or nootropic drugs/of antipsychotic or mood stabilizing drugs/of anticonvulsant or antispastic drugs (each yes/no). Analyses were conducted by SPSS v.22 (IBM, Chicago, IL, USA).

## 3. Results

### 3.1. Cohort Description

Through a chart review, we assembled a retrospective cohort of 46 patients treated with oxcarbazepine for behavioral disorders; of these, 12 were children, 18 adolescents, and 16 adults. The most frequent cause of brain injury was trauma (73.9%), and the median GCS at injury was 5.75. Lesion sites, as described by MRI, were heterogeneous: damage was reported for 58.7% patients in the frontal lobe, 32.6% in the callous body, 30.4% in the thalamus; 45.7% patients had diffuse axonal injury. Behavioral symptoms were also variable; 58.7% patients were irritable and reactive, 47.8% aggressive, 41.3% oppositional or non-compliant, 34.8% hyperactive/impulsive, 32.6% uninhibited and reckless, and 19.6% perseverant/confabulatory; 39.1% of patients had mood disorders. Considering drug therapy, oxcarbazepine was used at a median daily dose of 975 mg; 30.4% patients used also antipsychotic/mood stabilizing drugs, 15.2% antidepressant or nootropic drugs, 13% antispastic drugs. A complete description of the demographic and clinical characteristics collected is shown in Table 1, in which the cohort is split among patients who either improved or not.

Following the clinician-rated CGI-I, 21 (45.7%) patients were rated as having improved (CGI-I 2 or less), 25 (54.3%) as not (13 slightly improved, 7 unchanged, 1 worsened). On the caregiver-rated behavioral scales, considering RCI values, eight patients (50%) had significant improvements, and eight (50%) showed no significant change (7 improved slightly, 1 worsened slightly).

### 3.2. Variables Associated with Improvement

On the whole dataset (*n* = 46), we performed a stepwise logistic regression analysis probing the association between clinician-rated improvement on the CGI-I and the clinical variables we collected. We found (model *p* = 0.005, pseudo-R^2^ = 0.40, lack of fit *p* = 0.591) two variables associated with non-efficacy: the presence of a frontal lobe damage, OR = 0.078 (95% C.I. 0.008–0.749, *p* = 0.027) and the presence of symptoms of irritability/reactivity OR= 0.094 (95% C.I. 0.010–0.893, *p* = 0.040). We analyzed further the subpopulation of patients who had a behavior scoring scale, either ABS or SDQ, filled out by caregivers or self-rated, respectively, both before and after the introduction of oxcarbazepine. On the RCIs from these data (*n* = 16), we could perform a linear regression analysis in search of associations between the reliable change index of the scale available for each patient and the clinical variables we collected. We observed (model *p* = 0.001, R^2^ = 0.25) that a higher DRS at baseline (β = −0.356, *p* = 0.010) and a higher patient age (β = −0.327, *p* = 0.017) were associated with a lower RCI (which means more improvement).

### 3.3. Adverse Effects

Eight patients (17.4%) experienced adverse reactions considered by the treating physicians as possibly related to the use of oxcarbazepine: hyponatremia (3 patients; 6.5%), drowsiness or asthenia (2; 4.3%), psychiatric reactions (2; 4.3%), and one allergic reaction (2.1%). All adverse reactions were resolved at the time of discharge from rehabilitation, which required oxcarbazepine withdrawal in four cases and dose reduction in two; in two cases, the ADR was resolved by adding drugs or supplements. Details of ADRs are reported in Table 2.

## 4. Discussion

Carbamazepine and oxcarbazepine have been demonstrated as variably efficacious for the treatment of behavioral disorders in the context of psychiatric and neurological illness [25,37,38,39]. Given its lower potential for drug-drug interactions, we are using oxcarbazepine on patients in rehabilitation from brain injury in our clinical routine. Since the efficacy of oxcarbazepine has not been systematically assessed in brain-injured patients [7], any information on its effects, or on factors that may modulate them, is highly valuable from a clinical perspective. Regarding the composition of our study sample, there were predominantly pediatric patients; the majority of them had suffered brain trauma. In the literature, there is no clear indication of the efficacy of oxcarbazepine in such a population, with an exception made for some case-reports; thus, it is hard to compare our results with previous work.

Following clinician ratings and parent/self-ratings alike, around half of the patients who received oxcarbazepine improved their behavioral disorders. We could find significant roles for some clinical variables that were associated with the responder status.

Regarding clinician-rated CGI-I, we observed that the presence of a frontal lobe damage diagnosed by MRI and the presence of symptoms of irritability/reactivity were associated with non-response. Damage to the frontal lobe may indicate conditions that are connected with a “frontal lobe syndrome”, a generic definition for an organic behavioral disorder characterized by thought disorganization and personality changes including impulsivity, disinhibition, and aggression [31,40]. Symptoms of irritability and hyper-reactivity are the primary target of antipsychotic therapies used for the control of behavior, especially in young patients [41]; thus, it is not surprising that oxcarbazepine may be scantly efficacious at treating them, given its particular mechanism of action of suppressing neuronal hyperactivation. In fact, oxcarbazepine has been often used for mania, aggression, and mood swings in the context of bipolar disorders [23]. Of note, however, the presence of aggressive symptoms in our patients was not a factor associated with oxcarbazepine efficacy. It may be speculated that the presence of damage to the frontal lobe resembles a cognitive impairment; whether primary or degenerative, cognitive impairment is frequently the base of inappropriate comprehension of the environment (and relation with the environment) leading to frustration and irritability and eventually to abnormal behavior. Oxcarbazepine’s efficacy does not include cognitive improvement. On the caregiver/self-rated behavioral scales, higher DRS at baseline and higher patient age were associated with larger improvement. These results can be connected with a biased subjective perception of improvement. Indeed, parents of patients who are more severe at admittance to rehabilitation usually tend to overestimate their recovery; this bias effect can be even bigger when patients are asked to self-evaluate. Indeed, a qualitative result that emerged from several clinical records was that patients had the incapacity of perceiving their own behavioral issues and disability, possibly as part of an organic psychiatric illness. Indeed, a minority of patients reported complaints about their status, and they were those who developed depressive mood disorders after brain injury. The role of age may thus be involved with the fact that older patients are administered self-evaluations, with more positive bias, while children are evaluated by caregivers, with less bias. Moreover, age has been previously associated with the severity of behavioral disturbances in children [42]; following results of the present study, it may be hypothesized that younger patients incur deeper personality and behavior changes, as compared to older patients. Another important role of patients’ age regards the metabolism of oxcarbazepine. In the liver, oxcarbazepine is metabolized to its 10-monohydroxymetabolite, which is mostly responsible for psychopharmacological effects. The clearance of this metabolite decreases from childhood to adulthood. According to product labels for oxcarbazepine, 10-monohydroxymetabolite exposure in children up to 12 years is on average 40% higher than that of adults.

Of note, the response was not associated with any variable regarding pharmacological treatment: this is an expected finding in the context of a non-interventional retrospective study. In support of this interpretation, the dose of oxcarbazepine at titration was lower in the responder group as compared to the non-responder group. This also suggests that if a response could be achieved, it was achieved with oxcarbazepine doses of around 1 g/day. In general, the need for oxcarbazepine doses higher than 1–1.2 g/day for this use might be considered as a proxy of non-response. However, it is impossible to distinguish in a retrospective work the beneficial effects of oxcarbazepine from those of rehabilitation therapy, including cognitive-behavioral therapy that was administered to all patients in the study. Randomized, double-blind studies should be conducted to demonstrate the efficacy of oxcarbazepine in the treatment of behavioral disorders in brain-injured patients. Such studies are difficult to perform due to the heterogeneity of patients’ characteristics, involving different degrees of physical injuries, as well as sensory, motor, and cognitive disturbances and language disorders, all aspects that affect profoundly social interactions and behavior, together with organic psychiatric symptoms. Furthermore, patients recovering from brain injury usually require multiple drug therapies that must be tailored, rendering a standard clinical trial hardly feasible. In this view, more observational studies, possibly prospective, may also provide useful insight. The lack of high-quality studies on behavioral disorders following brain injury currently limits any conclusion on the efficacy of proposed drug therapies [7]. Clinical decisions should thus be based on safety and lack of drug-drug interactions, properties for which oxcarbazepine would be a suitable candidate [15]. In the present study, we have in fact found a significant occurrence of adverse drug reactions, including hyponatremia (known to occur [43]), as well as asthenia and psychiatric reactions. These adverse reactions should not be underestimated as they can compromise the precarious course of recovery of patients undergoing rehabilitation from brain injury. In particular, asthenia may impair physical rehabilitation, and psychiatric reactions may be mistaken for genuine psychiatric disorders requiring drug treatment. Future prospective studies should perform a thorough evaluation of adverse reactions to oxcarbazepine.

The generalizability of our results is limited by the composition of our study sample, which was small and predominantly of pediatric patients and brain traumatized patients. Whereas patients’ age may be an important factor limiting generalizability, the traumatic etiology would be less limiting as it is the most frequent cause of brain injury. A technical limitation of our work regards the grouping of symptoms that we chose post-hoc; however, we tried to follow groups of symptoms that were previously described for the frontal syndrome [29] and that were included in the DSM-IV-TR classifications for conduct disorder and oppositional defiant disorder. We explored different groupings for symptoms and found irritability/over-reactivity was a significant factor only when taken on its own. Another limitation regards the limited sample size of the caregiver/self-evaluation sample, due to the absence of scoring scales in the clinical charts. In addition, the caregiver/self-evaluation questionnaires were not entirely equivalent and interchangeable. Future studies should thoroughly implement behavior rating scales such as the ABS and administer them only to caregivers, to minimize bias. In addition, the consistency between clinician- and caregiver-rated efficacy judgments should be verified, a task that we could not carry out due to the low sample size.

## 5. Conclusions

We have investigated retrospectively which clinical factors may be associated with responses to oxcarbazepine in behavioral disorders, in the setting of rehabilitation from brain injury. We found possible roles for the presence of frontal lobe damage, of a diagnosis of irritability/hyper-reactivity, of lower patient age, and of lower disability ratings at baseline, in association with non-response. Given the nature of this study, it was expected that pharmacological factors could be not associated with the responder status. Future studies should implement experimental protocols or at least prospective observational protocols, in order to investigate the efficacy of oxcarbazepine. Such studies may be hard to conduct, given the need of patients for multiple drug, physical, and psychological therapies.

## Figures and Tables

**Table 1 brainsci-11-00949-t001:** Description of demographic and clinical characteristics of the study sample.

			Not Improved on CGI-I (*n* = 21)	Improved on CGI-I (*n* = 25)	*p* Value *
Clinical characteristics	Brain injury etiology	Trauma	14	20	0.059
		Stroke	1	4	
		Anoxia	4	0	
		Other	2	1	
	Age (years)	Median	17.2	17.4	0.589
		1st–3rd q	12.8–20.7	15.0–26.6	
	GCS at injury	Median	5.75	5.75	0.761
		1st–3rd q	5–6.5	4.5–6.5	
	GOS at admittance to rehabilitation	Median	3	3	0.789
		1st–3rd q	3–4	3–3.5	
	Coma duration (days)	Median	0	0	0.366
		1st–3rd q	0–16	0–9	
Brain damage as seen through MRI	Frontal lobe	Yes	15	12	0.108
		No	6	13	
	Callous body	Yes	6	9	0.592
		No	15	16	
	Thalamus	Yes	5	9	0.371
		No	16	16	
	Diffuse axonal injury	Yes	10	11	0.806
		No	11	14	
Behavioral symptoms at baseline	Oppositional/noncompliant	Yes	9	10	0.845
		No	12	15	
	Hyperactive/impulsive	Yes	7	9	0.850
		No	14	16	
	Uninhibited/reckless	Yes	8	7	0.467
		No	13	18	
	Confabulatory/perseverative	Yes	4	5	0.935
		No	17	20	
	Irritable/reactive	Yes	13	14	0.685
		No	8	11	
	Aggressive	Yes	11	11	0.571
		No	10	14	
	Disordered mood	Yes	7	11	0.460
		No	14	14	
Drug treatment	Oxcarbazepine daily dose (mg)	Median	1200	900	0.238
		1st–3rd q	725–1650	675–1200	
	Antidepressant/nootropic	Yes	3	4	0.872
		No	18	21	
	Antipsychotic/other mood stabilizer	Yes	8	6	0.301
		No	13	19	
	Antispastic	Yes	2	4	0.673
		No	19	21	
Disability and rehabilitation aspects	DRS at admittance to rehabilitation	Median	16.25	16.25	0.798
		1st–3rd q	11–22	14.5–20.5	
	DRS after rehabilitation	Median	5.5	5.5	0.247
		1st–3rd q	4.5–7	3–6	
	FIM social subscale at admittance to rehabilitation	Median	3	3	0.841
		1st–3rd q	1–5	1–5	
	FIM social subscale after rehabilitation	Median	3	5	0.482
		1st–3rd q	2.5–6.5	3–6	

**Legend.** 1st–3rd q: first and third quartiles; GCS: Glasgow Coma Score; GOS: Glasgow Outcome Score; MRI: Magnetic Resonance Imaging; DRS: Disability Rating Scale; FIM: Functional Independence Measure. * *p*-values reported regard: for categorical variables, Chi-square and/or Fisher’s exact tests, as applicable; for continuous variables, Mann–Whitney U tests.

**Table 2 brainsci-11-00949-t002:** Description of adverse events attributed to oxcarbazepine.

Sex	Age	ADR	Dose (mg)	Management	Outcome
M	29.4	Allergic reaction	Not titrated	Oxcarbazepine withdrawal	Resolved
M	18.5	Drowsiness	1050 BID	Dose reduction (750 BID)	Resolved
M	37.4	Severe asthenia	450 BID	Dose reduction, then withdrawal	Resolved only by withdrawal
M	18.6	Hyponatremia (134 mEq/L)	600 TID	Sodium supplementation	Resolved
F	20.9	Hyponatremia (132 mEq/L)	600 TID	Dose reduction (300 BID + 600)	Resolved
F	27.5	Severe hyponatremia (117 mEq/L), seizures	600 BID	Hospitalization, oxcarbazepine withdrawal, antiepileptic prophylaxis	Resolved
M	17.2	Psychotic behavior, derealization, mutism	300 BID	Risperidone add-on	Improved
M	8.8	Behavior worsened (increased agitation, aggression, screaming)	300 BID	Oxcarbazepine withdrawn, re-administered causing the same ADR, then permanently withdrawn	Resolved

## Data Availability

Data are available from the corresponding author upon request.

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
