# Peer review of "Oxcarbazepine for Behavioral Disorders after Brain Injury: Factors Influencing Efficacy"

_brainsci, 2021, doi:10.3390/brainsci11070949_

Round 1

Reviewer 1 Report

The authors provide a very interesting and well-written examination of oxcarbazepine for post TBI behavioural disorders. Given the difficulty in conducted RCTs, retrospective studies such as these provide important clinical information of prescribers. The authors have provided an excellent discussion of their findings and implications, as well as limitation to their study design and suggestions for future research. 

Introduction

·         It would be of benefit to include some brief statements regarding the mechanism of action of both drugs.

·         PG2 LN 51-52: Could more information be provided about the drug interactions with carbamazepine, and carbamazepine toxicity – including citations for statements made. Is there evidence from ABI samples regarding poor tolerance of carbamazepine?

·         PG 2 LN 53: Further information is required to explain why oxcarbazepine may have less drug interaction potential?

·         PG 2 LN 53: Could further information be added regarding evidence of better tolerance of oxcarbazepine than carbamazepine? Is this in ABI samples?

·         PG 2 LN 67: A citation for the Cochrane review is needed  

·         There are studies of carbamazepine use in TBI that should be added to the introduction, for example –

o   Azouvi P, Jokic C, Attal N, Denys P, Markabi S, Bussel B. Carbamazepine inagitation and aggressive behaviour following severe closed-head injury: resultsof an open trial.Brain Inj. (1999) 13:797–804. doi: 10.1080/026990599121188

o   Patterson  JF.  Carbamazepine  for  assaultive  patients  with  organicbrain   disease:   an   open   pilot   study.Psychosomatics.   (1987)28:579–81. doi: 10.1016/S0033-3182(87)72456-6

·         Are there any studies of oxcarbazepine in ABI samples? Could information regarding these studies also be included?

·         PG 2LN 68: Does this sentence refer to level of evidence in TBI samples specifically? If so – could this be clarified?

·         Given the majority of the sample were paediatric, a paragraph should be added regarding the evidence to date for these drugs in paediatric samples, specifically including both efficacy and harms

Method

·         PG 2 LN 79: please define ‘sABI’ unit – does the s stand for severe?

·         PG 2 LN82: the abbreviation CGI needs to spelled out. To understand the types of behavioural issues in your sample it would be helpful to include more information about the CGI measure and any relevant cut-offs used

·         PG 2 LN 83: how was the diagnosis of behavioural issues made?

·         PG 3 LN 97: which psychological assessment measures were used?

Discussion

·         PG 4 LN 208: could the authors elaborate on the mechanism of action for oxcarbazepine and why this mechanism of action may mean that it is not as effective for those with frontal lobe damage?

Author Response

Comments and Suggestions for Authors

The authors provide a very interesting and well-written examination of oxcarbazepine for post TBI behavioural disorders. Given the difficulty in conducted RCTs, retrospective studies such as these provide important clinical information of prescribers. The authors have provided an excellent discussion of their findings and implications, as well as limitation to their study design and suggestions for future research.

We thank the Reviewer for the consideration of our work and the useful suggestions and remarks made. We have answered to all comments in the point by point reply below.

Introduction

  • It would be of benefit to include some brief statements regarding the mechanism of action of both drugs.

We have added a sentence as suggested, on page 2 line 68: “Oxcarbazepine is a derivative of carbamazepine that has the same main mechanism of action, as they both prolong the inactivation phase of voltage-gated sodium channels.”

  • PG2 LN 51-52: Could more information be provided about the drug interactions with carbamazepine, and carbamazepine toxicity – including citations for statements made. Is there evidence from ABI samples regarding poor tolerance of carbamazepine?

We recognize that the word “toxicity” does not clearly indicate the adverse effects of carbamazepine or oxcarbazepine, as organ toxicity is a very uncommon event with both medications. To prevent this misunderstanding, we have now mentioned “adverse reactions” instead of toxicity on page 2, line 60. In addition, we have added a sentence, on page 2 line 60, regarding carbamazepine interactions and adverse reactions and the particular problems it may cause in patients with brain injury, as follows: “Regarding interactions, carbamazepine is a strong inducer of CYP3A4 and an inducer of glucuronyltransferase. In the clinical setting of brain injury rehabilitation, typical concomitant therapies include drugs with narrow therapeutic index metabolized by the two above enzymes, such as antibacterial chemotherapies, oral anticoagulants, drugs used for cardiovascular protection, other anticonvulsants and glucocorticoids. Adverse reactions to carbamazepine itself include hyponatremia, cognitive impairment, hepatic toxicity, anemia, and immune system activation with potentially fatal outcomes.”

  • PG 2 LN 53: Further information is required to explain why oxcarbazepine may have less drug interaction potential?

We have added the following sentence on page 2 line 70, to better explain the advantage of oxcarbazepine: “However, oxcarbazepine is a weak inducer of CYP3A4 and of glucuronyltransferase, such that the concomitant therapies typical of patients with sABI would not need dose adjustment, which represents a considerable added therapeutic value.”

  • PG 2 LN 53: Could further information be added regarding evidence of better tolerance of oxcarbazepine than carbamazepine? Is this in ABI samples?

We have added a sentence to clarify this aspect, as requested, on page 2 line 73: “The adverse reactions profile of carbamazepine and oxcarbazepine is almost identical. Better tolerance of oxcarbazepine stems from its increased manageability when drug combinations are frequent and inevitable, as in the case of patients with sABI. A lessened impact on the alterations of plasma levels of other drugs directly leads to reduced occurrence of adverse reactions and indirectly translates into better tolerance of oxcarbazepine and easier clinical management. “

  • PG 2 LN 67: A citation for the Cochrane review is needed

We thank the reviewer for pointing out the missing citation, which we have now added.

  • There are studies of carbamazepine use in TBI that should be added to the introduction, for example –

o   Azouvi P, Jokic C, Attal N, Denys P, Markabi S, Bussel B. Carbamazepine inagitation and aggressive behaviour following severe closed-head injury: resultsof an open trial.Brain Inj. (1999) 13:797–804. doi: 10.1080/026990599121188

o   Patterson  JF.  Carbamazepine  for  assaultive  patients  with  organicbrain   disease:   an   open   pilot   study.Psychosomatics.   (1987)28:579–81. doi: 10.1016/S0033-3182(87)72456-6

We thank the Reviewer for suggesting useful literature, which we have now cited on page 2 line 56.

  • Are there any studies of oxcarbazepine in ABI samples? Could information regarding these studies also be included?

To the best of our knowledge, there are no studies regarding oxcarbazepine used for behavioral disorders in the context of brain injury. However, a randomized controlled trial of carbamazepine in this exact context was published during the current peer review process. We have now cited this novel study on page 2 line 96: “A recent randomized controlled trial demonstrated that carbamazepine was efficacious in controlling aggression after traumatic brain injury, adding higher quality evidence to the field.”

  • PG 2LN 68: Does this sentence refer to level of evidence in TBI samples specifically? If so – could this be clarified?

This sentence was not referred to TBI; it was referred to patients with dementia. We have now clarified this sentence on page 2 line 87: “a recent meta-analysis on agitation due to dementia found no efficacy of oxcarbazepine”.

  • Given the majority of the sample were paediatric, a paragraph should be added regarding the evidence to date for these drugs in paediatric samples, specifically including both efficacy and harms

We agree with the Reviewer that a pediatric focus would be very important. However, to date there is no published work regarding pediatric patients with brain injury treated with carbamazepine and its derivatives for behavior disorders. We have now added a sentence on page 2 line 99 on this issue: “Studies with a pediatric focus should also be conducted, as there is currently no evidence base on the use of carbamazepine and its derivatives for the control of behavior after brain injury in pediatric patients.”

Method

  • PG 2 LN 79: please define ‘sABI’ unit – does the s stand for severe?

Yes. We have now defined the abbreviation sABI on page 1 line 1 after its first use.

  • PG 2 LN82: the abbreviation CGI needs to spelled out. To understand the types of behavioural issues in your sample it would be helpful to include more information about the CGI measure and any relevant cut-offs used

We agree with the Reviewer that an adequate explanation was lacking. We have now inserted a paragraph on page 3 line 121: “having a Clinical Global Impression -Improvement (CGI-I) rating of behavioral symptoms, (including but not limited to irritability, agitation, escaping, disruption, aggression) by which improvement is defined as CGI-I less than 3 on a scale of 1 (extremely improved) to 7 (extremely worsened)”.

  • PG 2 LN 83: how was the diagnosis of behavioural issues made?

We have now clarified this on page 3 line 116: “This diagnosis followed the presence of the criteria that by the DSM-IV-TR lead to the diagnosis of conduct disorder or oppositional-defiant disorder. However, due to the presence of a brain injury not fully recovered, a formal diagnosis of conduct disorder or oppositional-defiant disorder was not made until discharge, the details of behavioral disorders were reported on clinical files and monitored.”

  • PG 3 LN 97: which psychological assessment measures were used?

Since the severity and consequences of sABI can be variable across patients, the clinical protocol for psychological assessment can be more or less intense following the individual capability of patients.  Where appropriate, instruments used include: Wechsler Intelligence Scale for Children, Leiter-R, Vineland Adaptive Behavior Scales, Child Behavior Checklists, Adult Behavior Checklist, Strenghts and Difficulties Questionnaire for pediatric and adult patients, Agitated Behavior Scale. We have now clarified this on page 3 line 139: “From results of psychological assessments (including, where applicable: Wechsler Intelligence Scale for Children, Leiter-R, Vineland Adaptive Behavior Scales, Child Behavior Checklists, Adult Behavior Checklist), we recorded…”

Discussion

  • PG 4 LN 208: could the authors elaborate on the mechanism of action for oxcarbazepine and why this mechanism of action may mean that it is not as effective for those with frontal lobe damage?

As suggested, we have elaborated this hypothesis on page 8 line 270: “It may be speculated that the presence of damage to the frontal lobe resembles a cognitive impairment; whether primary or degenerative, cognitive impairment is frequently the base of inappropriate comprehension of the environment (and relation with the environment) leading to frustration and irritability and eventually to abnormal behavior. Oxcarbazepine’s efficacy does not include cognitive improvement.“

Reviewer 2 Report

It is a remarkable paper about the efficacy of oxcarbazepine for behavioral disorders after brain injury. However, several revisions are required as follows:

  1. The ethical approval should be described.
  2. Overall severity of brain damage should be included in factors influencing efficacy. The study has covered the several factors influencing the efficacy of oxcarbazepine.
  3. The potential influences of other psychotropic drugs on the efficacy of oxcarbazepine for brain disorder after brain injury.
  4. The target symptoms of oxcarbazepine is unclearly defined. Thus, the study goal is unclear and arbitrary.
  5. The sample size is so small. The content should be described in limitations.
  6. The detailed diagnoses for the study participants should be described. (DSM or ICD)

Author Response

Comments and Suggestions for Authors

It is a remarkable paper about the efficacy of oxcarbazepine for behavioral disorders after brain injury. However, several revisions are required as follows:

We thank the Reviewer for the consideration of our work and for providing useful suggestions and requests for improvements. We have answered to all comments in the point by point reply below.

The ethical approval should be described.

Following the Italian law, retrospective studies do not require ethical approval, because all clinical data are already collected under ethical standards and at admittance patients sign an authorization for the use of their clinical data in anonymized form to conduct any clinical research. The statement is reported on page 9 line 359.

Overall severity of brain damage should be included in factors influencing efficacy. The study has covered the several factors influencing the efficacy of oxcarbazepine.

The severity of brain damage (GCS) was used as variable in all models but it did not result to be significantly associated with the outcomes. To clarify our analyses, we have now stated in the methods section the full list of variables used, on page 4 line 178: “The full list of variables tested for association was: Glasgow coma score (GCS) at injury; duration of coma (days); Glasgow outcome score (GOS) at admittance to rehabilitation; Disability rating scale (DRS) at admittance to rehabilitation and at discharge; Functional independence measure (FIM) at admittance to rehabilitation and at discharge; age at start of oxcarbazepine administration (years); oxcarbazepine titrated dose administered (mg/day); damage evidenced by MRI to the frontal lobe / callous body / thalamus / diffuse axonal damage (each yes/no); presence at neuropsychological evaluation of opposition, noncompliance / of hyperactivity, impulsivity / of uninhibition, recklessness / of confabulation, perseveration / of irritability, reactivity / of aggression / of disordered mood (each yes/no); concomitant use of antidepressant or nootropic drugs / of antipsychotic or mood stabilizing drugs / of anticonvulsant or antispastic drugs (each yes/no).”

The potential influences of other psychotropic drugs on the efficacy of oxcarbazepine for brain disorder after brain injury.

We have now clarified in the methods (page 4 line 178) the list of variables investigated, which included the use of other psychotropic drugs and of other drugs that could have a stabilizing or sedating or hypnotic effect.

The target symptoms of oxcarbazepine is unclearly defined. Thus, the study goal is unclear and arbitrary.

We have now specified the target symptoms of oxcarbazepine and the study objectives at the end of the introduction section, page 2 line 104: “The objectives of this study were 1- to describe the efficacy of oxcarbazepine on the containment of symptoms of conduct disorder and oppositional defiant disorder in our clinical population; 2- to verify whether oxcarbazepine efficacy was associated with differences in any clinical variables.”

The sample size is so small. The content should be described in limitations.

We have now acknowledged the small sample size of this study at the end of the discussion section, page 9 line 324: “The generalizability of our results is limited by the composition of our study sample, which was small and predominantly of pediatric patients and brain traumatized patients.”

The detailed diagnoses for the study participants should be described. (DSM or ICD)

We have now clarified in the methods section (page3 line 116) the diagnostic procedure and criteria used. We classified patients’ symptoms based on the items in DSM-IV-TR categories of conduct disorder and oppositional defiant disorder. However, since these were patients with brain injury which was not yet recovered, we did not formulate a definite diagnosis, we rather described the symptoms present for each patient and followed these symptoms during the course of neurological recovery. At the end of rehabilitation, patients with residual behavioral symptoms enough to support a DSM-compliant diagnosis were discharged with the appropriate diagnosis.

Round 2

Reviewer 2 Report

Although the authors have revised the manuscript according to reviewers' comments, the research problems have not been alleviated.